# Transformers Can Do Arithmetic with the Right Embeddings

**Sean McLeish**[1*], **Arpit Bansal**[1*], **Alex Stein**[1], **Neel Jain**[1], **John Kirchenbauer**[1],
**Brian R. Bartoldson**[2], **Bhavya Kailkhura**[2], **Abhinav Bhatele**[1], **Jonas Geiping**[3],
**Avi Schwarzschild**[4], **Tom Goldstein**[1]
[1] University of Maryland, [2] Lawrence Livermore National Laboratory, [3] ELLIS Institute Tübingen,
Max Planck Institute for Intelligent Systems, Tübingen AI Center, [4] Carnegie Mellon University

## Abstract

The poor performance of transformers on arithmetic tasks seems to stem in large part from their inability to keep track of the exact position of each digit inside of a large span of digits. We mend this problem by adding an embedding to each digit that encodes its position relative to the start of the number. In addition to the boost these embeddings provide on their own, we show that this fix enables architectural modifications such as input injection and recurrent layers to improve performance even further.

With positions resolved, we can study the logical extrapolation ability of transformers. Can they solve arithmetic problems that are larger and more complex than those in their training data? We find that training on only 20 digit numbers with a single GPU for one day, we can reach state-of-the-art performance, achieving up to 99% accuracy on 100 digit addition problems. Finally, we show that these gains in numeracy also unlock improvements on other multi-step reasoning tasks including sorting and multiplication. [2]

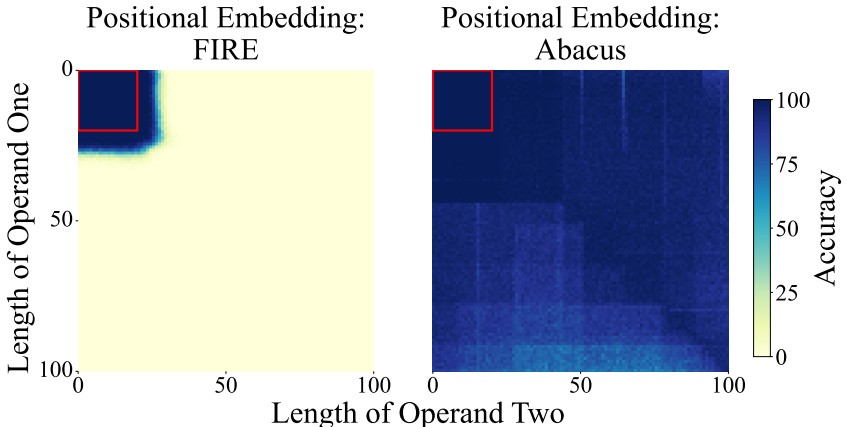

Figure 1: Zero shot exact match accuracy on addition using depth sixteen transformer (decoder only) models trained on operands of up to 20 digits. Compared to state-of-the-art embeddings (left), our new *Abacus Embeddings* (right) dramatically improve generalization to unseen digit lengths. The interior of the red square denotes the training distribution. Accuracies are averaged over three trials.

---

[*] Equal Contribution, correspondence to: `smcleish@umd.edu`, `bansal01@umd.edu`.
[2] Code available on GitHub: github.com/mcleish7/arithmetic.

38th Conference on Neural Information Processing Systems (NeurIPS 2024).

# 1  Introduction

Much of the recent work on Large Language Models (LLMs) focuses on their ability to solve problems in natural language and code generation. Despite progress in these domains, transformers still struggle to perform complex multi-step and algorithmic reasoning tasks in a zero shot setting without resorting to tool use. To study algorithmic reasoning in a sterile laboratory setting, the academic community focuses on simple arithmetic test problems like addition. Addition is simple enough that modest-sized LLMs can (in principle) be trained from scratch to do it without running into capacity and training budget limitations, yet complex enough that even large industrial models fail on large numbers without a code interpreter [Loeber, 2024].

Training transformers for arithmetic enables us to study several important questions. First, we ask what architectural design choices, dataset characteristics, and training pipeline variants are required to learn a many-step reasoning process like multi-digit addition? Going deeper, we then investigate whether these models are capable of *logical extrapolation*—can they solve problems of greater size and difficulty than those that appear in their training set?

Prior studies indicate that addition is hard for transformers [Lee et al., 2023, Shen et al., 2023, Zhou et al., 2023, 2024]. Our experiments indicate that this difficulty stems from their inability to clearly represent the exact position of a digit within a long sequence of digits. To address this problem, we propose a simple modification to the data representation that directly addresses this shortcoming. Our *Abacus Embeddings* are simple learned positional embeddings that are used to encode positions within each span of numerical tokens. Combining Abacus Embeddings and standard positional embeddings, we observe dramatic improvements in accuracy such that models trained with at most 20 digit operands can generalize to problems with 120 digit operands. This represents a state-of-the-art generalization factor of $6\times$, with the previous state of the art being only $2.5\times$. To the best of our knowledge, these are the longest sequences on which learned addition has ever been demonstrated.

We also study several other methods of improving arithmetic and generalization in transformers. We find that incorporating *input injection*—skip connections inserted between the input layer and each decoder layer—can reduce generalization errors by 50% over the Abacus Embedding baseline. We also find that together with our embeddings looped transformer architectures, which contain recurrent layers in which the same parameters are re-used multiple times, can achieve near-perfect generalization on addition problems we consider.

Since our proposed methods solve large addition problems successfully, we evaluate whether the same approaches can be used to improve other kinds of algorithmic learning. We explore multiplication problems of up to 15 digit numbers and sorting over arrays of up to 10 numbers, making this the first study of extreme length generalization techniques for addition that transfer to other algorithmic tasks. Our contributions can be summarized as follows.

- We propose a new positional embedding called *Abacus Embeddings* to better capture the significance of each digit, which leads to near-perfect in-distribution generalization.
- We show that when we combine Abacus Embeddings with input injection and looped transformers performance further improves, increasing from $92.9\%$ to $99.1\%$ in out of distribution accuracy, an $87\%$ reduction in error compared to using the embeddings with standard architectures alone.
- We push length generalization beyond existing work and show that our models can solve problems with six times as many digits as the largest samples in the training set, whereas the previous state of the art is only two and a half times.
- We extend our findings to more complex problems including multiplication and sorting where we show length generalization in these domains.

# 2  Related Work

**Arithmetic and Algorithmic Reasoning.**  Solving arithmetic with next token prediction is a difficult problem that attracts a lot of attention [e.g. Saxton et al., 2019]. However, in zero-shot settings, even incredibly strong commercial API models struggle with very large addition problems (e.g. up to 100 digits) without access to tools. Among attempts to improve arithmetic performance of transformer-based models, reversing the digits so the arguments are written with the least significant

digit first is popular [Lee et al., 2023, Shen et al., 2023, Zhou et al., 2023, 2024]. Furthermore, changing the data format by adding explicit index characters improves model capability for addition [Zhou et al., 2023, 2024, Olsson et al., 2022]. Other work approaches arithmetic by embedding real numbers by scaling a single fixed token-embedding for numbers [Golkar et al., 2023]. Moreover, Dziri et al. [2023] show multiplication is a hard problem for GPT-3 [Brown et al., 2020] even when finetuned on this task. Dziri et al. [2023] further show that GPT-4 [OpenAI, 2023] struggles to obtain high in-distribution accuracy on multiplication, even with a scratchpad. However, Lee et al. [2023] find that with a detailed scratchpad, small transformers can perform multiplication in-distribution.

Arithmetic is a subset of the larger class of algorithmic reasoning problems that focus on the ability to learn and execute algorithms and generalize to longer problems [Anil et al., 2022b, Jelassi et al., 2023, Yang et al., 2023b, Veličković et al., 2022, Rodionov and Prokhorenkova, 2024, Testolin, 2024]. The more general algorithmic reasoning field includes work on various architectures and data modalities aimed at learning algorithms from data. Veličković et al. [2022] and Rodionov and Prokhorenkova [2024], for example, train neural networks to execute specific algorithmic tasks by training on input-output pairs as well as intermediate steps and hints. In a similar vein and although initially appreciated for efficiency, weight sharing and recurrence can be used to make models adaptive and help generalize to harder problems [Dehghani et al., 2018, Sukhbaatar et al., 2019, Lan et al., 2020, Ibarz et al., 2022]. Schwarzschild et al. [2021] and Bansal et al. [2022] explore an end-to-end learning approach using recurrent convolutional neural networks to learn algorithms from input-output pairs, tackling algorithmic tasks like prefix sums, mazes, and chess. Weight sharing for algorithmic reasoning is also helpful with transformers and we use the *looped transformer* in some of our experiments below. A looped transformer has a transformer block called recurrently on its own output lending itself to executing iterative algorithms [Giannou et al., 2023, Yang et al., 2023a, de Luca and Fountoulakis, 2024]. Additionally, recent work aims to improve reasoning in LLMs [Zhou et al., 2023], but McLeish et al. [2024] demonstrate that LLMs, even with code interpreters, are less than perfect at algorithmic reasoning tasks, indicating a crucial need for advancements in our methodologies. This paper takes a step towards improving LLM arithmetic and algorithmic capabilities without tool use.

**Positional Embeddings.** Indicating the position of tokens in a sequence to transformer models is critical for language modeling [Vaswani et al., 2017]. Absolute positional embeddings (APE) are learned embeddings that are added to token embeddings before the first layer of the transformer [Vaswani et al., 2017]. However, these absolute embeddings inhibit length generalization [Press et al., 2022]. To address this issue, Shaw et al. [2018] propose relative embeddings (RPE) which are embedded during the attention computation, a mechanism further simplified by Raffel et al. [2020]. Others build on these works to improve length generalization including Sandwich [Chi et al., 2023], Kerple [Chi et al., 2022], and Alibi [Press et al., 2022] positional embeddings. Additionally, Kazemnejad et al. [2023] show that decoder layers can still learn positional information with no explicit positional embeddings. No positional embeddings (NoPE) can achieve good length generalization performance for small algorithmic tasks and even outperform some specialized embeddings. Rotary Positional Embeddings (RoPE) [Su et al., 2024] are commonly used in state-of-the-art open source transformers [e.g. Touvron et al., 2023]. However, RoPE does limit the length generalization as models are trained only using rotations based on training data length [Kazemnejad et al., 2023, Press et al., 2022]. For improved length generalization, one can add post-training extensions [Peng et al., 2024]. The latest and most useful for arithmetic is Functional Interpolation for Relative Position Embeddings (FIRE) [Li et al., 2023]. FIRE shows the strongest length generalization to date, which leads to length generalization by $2.5\times$ on addition [Zhou et al., 2024] when combined with randomized embeddings [Ruoss et al., 2023]. We go into more detail on some of these positional embeddings in Appendix A.1.1. In this work, we focus on NoPE and FIRE embeddings since these are the best performers for addition in reversed format among existing embeddings [Zhou et al., 2024].

## 3  Achieving Length Generalization for Addition

We study a range of methods for improving the arithmetic capabilities of language models trained from scratch centering on two main hypotheses: (1) the positional information for individual digits within numbers is being lost and (2) recurrence can improve the reasoning abilities of transformer architectures on multi-step arithmetic reasoning problems. We briefly discuss the training and evaluation setup before describing each of our improvements in detail.

| Least Significant Digit First: | 1 | 2 | 3 | 4 | + | 1 | 2 | 3 | 4 | = | 2 | 4 | 6 | 8 |
|---|---|---|---|---|---|---|---|---|---|---|---|---|---|---|
| Most Significant Digit First: | 4 | 3 | 2 | 1 | + | 4 | 3 | 2 | 1 | = | 8 | 6 | 4 | 2 |
| Abacus Embeddings: | 1 | 2 | 3 | 4 | 0 | 1 | 2 | 3 | 4 | 0 | 1 | 2 | 3 | 4 |
| Absolute Embeddings: | 1 | 2 | 3 | 4 | 5 | 6 | 7 | 8 | 9 | 10 | 11 | 12 | 13 | 14 |

Figure 2: Visualization of data formats and positional embeddings. *Abacus Embeddings* give the same positional embeddings to all digits of the same significance.

**Experimental Setup.** We train decoder-only causal language models to solve addition problems. Following prior work [Zhou et al., 2023, 2024, Shen et al., 2023, Kazemnejad et al., 2023, Lee et al., 2023], inputs are formatted least significant digit first, e.g. $98282 + 3859172 = 2787472$. Unlike prior work, we do not add any padding between digits [Shen et al., 2023] and do not pad any numbers with zeros, neither in the case of carry digits [Zhou et al., 2024], nor to make all operands the same length [Shen et al., 2023]. We train on all combinations of operand lengths less than or equal to $i$ and $j$ where $i$ and $j$ are the maximum lengths of the first and second operands, respectively. For this study all training sets have 20 million samples and $i = j$, hence we can use one number to define the dataset $i$, where $i$ is the maximum length of either operand. We sample data with replacement and we stratify the data, so that all length pairs $(i, j)$ are equally sampled during training. To facilitate training of many models from scratch, we use a language model cramming setup [Geiping and Goldstein, 2023] and limit each training run to 8 exaFLOP of compute (a single Nvidia RTXA4000 GPU for 24 hours); for multiplication results we allow 64 exaFLOP (eight Nvidia RTXA4000 GPUs for 24 hours). During training, we mask the input question and only compute loss on the answer digits. For further details on data construction and training we refer to Appendix A.2.

We report model accuracy for each $(i, j)$ length pair and unlike most existing work, we also include accuracy for pairs where $i \neq j$ to highlight all instances of extrapolation. This extensive tabulation is costly and makes inference the main computational burden of this study. Since our training pipeline produces fairly consistent results, we report the mean over three runs (rather than using a best-of-ten reporting scheme [Zhou et al., 2024]). We measure accuracy in the strict sense where only exact matches of all output digits are counted as correct, i.e. if a single digit is incorrect then the example is marked as wrong and we refer to this as *exact match accuracy*. We have the following three evaluation categories: (i) in distribution (ID) where the models are tested on problems up to the maximum size seen during training; (ii) out of distribution (OOD) where the models are tested on problems greater than the maximum size seen during training but both operands are at most 100 digits; (iii) and extreme out of distribution (100+ digit OOD) where the models are tested on problems where both operands are of the same length and are both more than 100 digits and less than 160 digits. In the 100+ OOD setting, we only analyze problems where the operands are the same length ($i = j$) due to inference costs at this scale.

We consider two standard transformer architectures. First, we use a standard autoregressive transformer model where multiple decoder layers are stacked in a feedforward manner. Second, we enhance this standard transformer model by incorporating *input injection*, where the embedded inputs are added to the input of each decoder layer [Ma et al., 2022, Bansal et al., 2022, Anil et al., 2022a]. We visually describe the architectures in the Appendix Figure 22.

## 3.1 Abacus Embeddings Help Align Digits

From prior work and our own initial experiments, we observe that even when input numbers are presented least-significant digit first and training data is stratified and abundant (several million examples), standard transformers struggle to learn multi-digit addition. We also observe that humans do long addition by first aligning the digits of the same significance into columns. Thus, our first hypothesis is that the significance of each digit (i.e. each digit's position relative to the beginning of the number) is not easy for transformers to represent, and that this sub-problem presents more of a hurdle than the actual addition itself.

Prior work addresses this by proposing explicit index hints in the inputs and outputs of the addition, for example $a6b7c5 + a1b6c3 = a7b3c9$, finding that transformers perform much better on addition with the information provided by such hints [Zhou et al., 2023, 2024]. However, index hints of this form increase the input context length required and *double* the output length and inference cost of solving a given addition problem. Furthermore, Zhou et al. [2024] find that the ability of models trained with index hints to generalize is sensitive to the particular random initialization. Zhou et al.

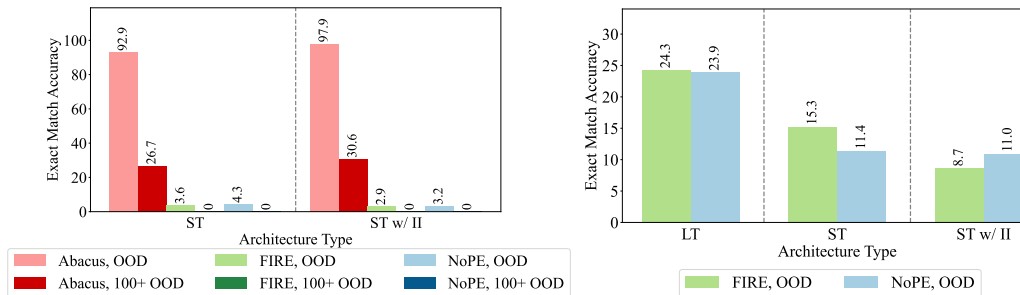

Figure 3: **Left:** Mean exact match accuracy of three models of depth sixteen on size 20 data, varying the architecture and embeddings. Abacus Embeddings improve accuracy for addition over FIRE and NoPE Embeddings. **Right:** Mean exact match accuracy of three models of effective depth sixteen on size 40 data, varying over NoPE or FIRE embeddings and architectures. Recurrent looped transformer models improve accuracy for addition for both the FIRE and NoPE embeddings.
*Looped transformer (LT):* Weight tied decoder layers, with input injection and progressive loss. *Standard Transformer (ST):* Stacked decoder only layers. *Standard Transformer with Input Injection (ST w/ II):* Standard Transformer with input features added to the hidden representation between each decoder layer.

[2024] highlight this by training models with different random seeds, varying weight initialization and data input order seeds, showing the variance in the performance of these models can vary from near perfect on 100 digit addition to 0% accuracy at 90 digit addition.

To address the limitations of transformers at representing positional information, we design a specially built positional embedding that encodes the location of each digit relative to the start of the current number. We call this *Abacus Embeddings*. We apply the same positional embedding to all digits of the same significance, providing an explicit signal that the model can use to align digits. We visually describe these embeddings in Figure 2.[3]

We take inspiration from *Randomized Embeddings* [Ruoss et al., 2023] but instead of using random ascending indices to represent positions in a sample, we use consecutive ascending indices with a random starting position to allow for length generalization. Specifically, during training we give consecutive positional embeddings to each digit in a number, starting from a randomly chosen offset value from $U[1, k]$, where $k$ is a hyperparameter. Unless otherwise stated the default value for $k$ in this study is 100 and show this can be varied in Appendix A.5. For example, if the input is 123, the positional encodings are $\beta, \beta + 1, \beta + 2$ where $\beta \sim U[1, 100]$, which are then passed through a learned embedding matrix. The value sampled from $U[1, k]$ is the same for all numbers in a batch, meaning all digits of the same significance obtain the same positional embedding. This training scheme allows the model to see a wide range of positional embeddings, even when training sequences are short. At test time, each positional embedding begins from one, i.e. $\beta = 1$.

**Abacus Embeddings Solve Addition.** Abacus Embeddings improve generalization performance up to 100 digits and beyond for standard transformer architectures. In Figure 3 (left), we highlight the comparative boost Abacus Embeddings have over standard transformer architectures and embeddings for performing addition, taking the mean accuracy of three models in all cases. The accuracy results for the standard transformer models trained with FIRE and Abacus, tested both in-domain (ID) and out-of-domain (OOD), are also shown in Figure 1. Additionally, in Appendix A.6, we present similar 2D grid plots for several other experiments that are depicted as bar charts in the main text. Zhou et al. [2024] find that operand lengths of up to forty digits are required during training for good generalization to 100 digit addition during testing (albeit not robustly). We find that with our Abacus Embeddings, we can achieve similar accuracy and larger extrapolation using a standard model with input injection trained on maximum operand sizes of 20 digits.

As Abacus Embeddings are a variant of absolute positional embeddings, technically they cannot generalize beyond the relative positions seen during training. However the hyperparameter $k$ that

---

[3]In Appendix A.3, we motivate these embeddings further with experiments demonstrating their utility in solving a bitwise OR task.

randomizes the starting offset used for each individual addition example can be increased to enable generalization by training a larger range of embeddings for a given computational budget. Relatedly, Appendix Figure 9 shows that training on larger datasets improves performance, even for operands with fewer than 100 digits.

## 3.2 Recurrence In Transformers Boosts Performance

With positional embeddings addressed, next we explore whether recurrent architectures can further improve the ability of transformers to perform multi-digit addition. We use the term *recurrent block* to refer to a set of decoder layers with distinct weights and *recurrences* to refer to the number of times the recurrent block is repeated. We use the term *effective depth* to mean the number of layers used in a transformer, whether their weights are unique or not. Unless otherwise stated, we use a maximally recurrent architecture, i.e. only one unique layer recurred to achieve the effective depth. We also employ input injection, skip-connections that propagate a copy of the input to each layer in the network.

**The Benefits of Recurrence.** In Figure 3 (right), we compare all architecture variants using both FIRE and NoPE embeddings trained on addition over operands with up to 40 digits. Despite having approximately $10\times$ fewer parameters than the other models, we see that the looped transformer (recurrent, with input injection and progressive loss), achieves the best out of distribution performance using either position embedding. In Figure 9 in the Appendix, we show this result is robust across multiple training data sizes.

With recurrent models, we can choose to vary the number of recurrences for each forward pass while training. This tends to improve generalization to harder tasks at test time and is also refered to as *progressive loss* computation [Bansal et al., 2022]. This loss function is a convex combination of the loss values from two forward passes, one with the nominal number of recurrences (so 16 for a $1 \times 16$ model) and one with a random smaller number of recurrences.

Next, we explore the effect of varying the size of the recurrent block while keeping the effective depth fixed. We perform this ablation by halving the number of layers in the recurrent block and doubling the number of recurrences, sweeping from a model with sixteen layers in the block and a single recurrence ($16 \times 1$, i.e. a standard transformer), through to one layer in the block but with sixteen recurrences ($1 \times 16$). Analyzing these results in Figure 4, we show further performance improvements are possible in some cases with the combination of both recurrence and Abacus Embeddings. In particular, a model with two recurrences ($8 \times 2$) incurs half the error of the purely non-recurrent model ($16 \times 1$) for OOD problems and enjoys increased accuracy on $100+$ OOD problems.

Finally, in Appendix A.7.3, we vary the effective depth of the models to analyze the impact of parameter count on this task, across Abacus, FIRE and NoPE embeddings. Although the experiments presented in Figure 4 are a fair comparison across depth, the purely standard transformer models have many more parameters than their recurrent counterparts. In Table 3 in the appendix, we record the parameter counts to the nearest million.

## 4  Pushing the Limits of Algorithmic Reasoning for Transformers

While there is an emphasis on addition as a difficult problem in existing work, our method's strong performance allows us to extend to even more difficult problems, including multiplication and sorting and even multiple operations at once.

### 4.1  Addition and Subtraction

We train models on a dataset made up of an even mix of addition and subtraction samples. In Figure 5, we show results from models with 8 layers in the recurrent block and 2 recurrences trained with exactly the same hyperparameters used to train the addition models above. We see that these small transformer models can simultaneously learn to extrapolate for both the symmetric operation of addition and the anti-symmetric operation of subtraction using Abacus Embeddings.

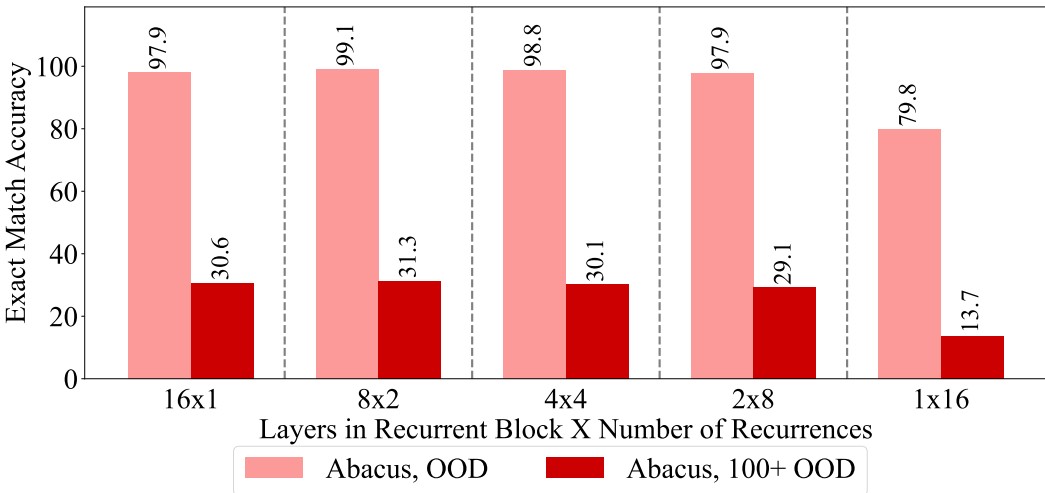

Figure 4: Varying the size of the recurrent block, while maintaining an effective depth of 16 and training on size 20 data. We see that a recurrent model with eight layers in the recurrent block and two recurrences is the most accurate of all effective depth 16 models, halving the error rate of a standard model with input injection in the OOD evaluation. (See Figure 17 for results with FIRE and NoPE.)

## 4.2 Integer Multiplication

We now study a harder task, multiplication of natural numbers, where the length of the output may be the sum of the lengths of the operands. Compared to addition, where the output is at most one digit more than the longest operand, multiplication has longer-distance dependency and the output length scales much faster as problem size increases.

To adapt from addition to multiplication, we make some small changes to our set-up. First, we remove the input injection from inside the recurrent block and second, we divide the gradients in the recurrent block by the number of recurrences, down-weighing the gradient update from batches with many recurrences [Bansal et al., 2022]. (We analyze the impact of these design decisions for addition models in Appendix Figure 19.) We only examine looped transformers as the compute required for training and hyperparameter search for multiplication is far greater than for addition, limiting us to a much smaller scale analysis.

Abacus Embeddings help looped transformers reach near-perfect accuracy in-distribution for multiplication. In Figure 6, we show how the training distribution, surrounded by the red square fully saturates with Abacus Embeddings. In fact, models with our Abacus Embeddings achieve higher in

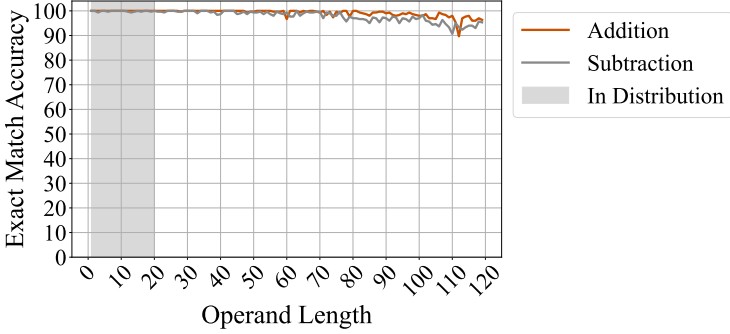

Figure 5: Models which have 8 layers in recurrent block and 2 recurrences, trained on size 20 addition and subtraction data, each line is the average of 3 models. We see that it is possible to have extreme generalization whilst learning multiple tasks.

distribution accuracy on 15 digit multiplication than prior work [Shen et al., 2023] and do not require padding each operand to the same length with zeros. In particular, we highlight that the specific problems that models trained with FIRE embeddings struggle to solve are the hardest problems in the training set and Abacus Embeddings outperform them in this key area (see the lower right corner of the red boxes in Figure 6).

## 4.3 Array Sorting

While both addition and multiplication accept only two operands, we now analyze the task of sorting arrays of multiple variable length numbers, a more challenging testbed for evaluating the generalization abilities of our Abacus Embeddings. We present each sorting problem using alphabetical indices for each (reversed) number in an input array where the expected output is the alphabetical indices in ascending order. For example, $a : 64957, b : 99963, c : 10218, d : 7141, e : 05781 = d, e, b, a, c$. We train with arrays of up to 10 numbers each having up to 10 digits and then evaluate with arrays of up to 30 numbers each having up to 30 digits. We give more detail on the sorting data construction process in Appendix A.2.

In this setting, we explore two axes of generalization. First, we increase the maximum possible length of the input numbers to 30 digits while maintaining the maximum array length to 10 and refer to this scenario as "OOD (number length - 30)." Second, we increase the number of inputs in the array to be sorted to 30 while keeping the maximum digit length of each number at 10 and term this scenario "OOD (array

Table 1: Exact match accuracy for sorting with various positional embeddings. All results are percentages of the test set and all models here are standard transformers with eight layers.

|  | FIRE | Abacus | Abacus + FIRE |
|---|---|---|---|
| OOD (number length - 30) | 55.32 | **68.63** | 67.28 |
| OOD (array length - 30) | **21.35** | 9.67 | 21.11 |
| All OOD (30 × 30) | 3.73 | 2.65 | **4.48** |
| All OOD (20 × 20) | 14.65 | 9.78 | **16.91** |

Table 2: Accuracy for sorting with various architectures for sorting. ST denotes standard transformer, ST w/ II denotes standard transformer with input injection, and LT denotes looped transformer models. The standard transformer has the best exact match accuracy. When measuring the accuracy on identifying only the minimum element of the array, looped transformers outperform all others. All results are percentages of the test set.

|  | ST | ST w/ II | LT |
|---|---|---|---|
| All OOD (exact string match) | **4.48** | 3.84 | 2.60 |
| All OOD (min. elem. only) | 49.73 | 60.09 | **68.51** |

length - 30)." Finally, we consider a scenario where both axes are increased simultaneously, referred to as "all OOD."

In Table 1, we illustrate the performance of a standard transformer (eight layers) trained with different embeddings—FIRE, Abacus, and their combination. Again, our results demonstrate that the combined embedding approach enhances the model's ability to generalize, surpassing the performance of either

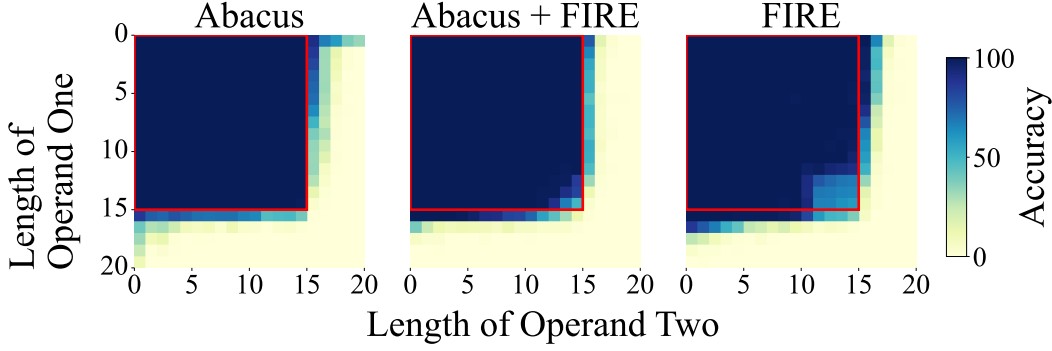

Figure 6: Exact match accuracy of looped transformer models trained on multiplication, with four layers in the recurrent block and four recurrences. The red square denotes in distribution testing on up to 15 digit operands. We see the models with Abacus Embeddings achieve near perfect in distribution accuracy. Combining Abacus Embeddings with FIRE also improves in distribution accuracy on the hardest in distribution problems (bottom right), comparing to the FIRE-only baseline.

embedding alone in the "all OOD" setting. However, in Table 2, we observe mixed results when pairing the Abacus+FIRE Embeddings combination with different model architectures with effective depth eight. For sorting, different architectures appear to be better suited to different types of extrapolation, for example the looped transformer is best at extrapolating for finding the minimum element but not for sorting the whole array.

Overall, the superior sorting performance of the Abacus Embeddings underscores their potential utility across a broader spectrum of algorithmic tasks beyond basic arithmetic. Abacus Embeddings may be instrumental in use cases requiring transformer models to perform a variety of complex positional, numerical, and/or relational reasoning tasks.

### 4.4 Abacus and Relative Embeddings

As Abacus Embeddings are only applied to numbers, to incorporate Abacus Embeddings into a general purpose model, they must be compatible with other relative embeddings to maintain good downstream performance on non-arithmetic tasks. We examine these types of combinations here and conclude that Abacus Embeddings complement techniques that are good for natural language well, suggesting that these combinations could be powerful for large-scale general models.

Although Abacus Embeddings are implicitly combined with NoPE (no positional embeddings) embeddings for all experiments seen so far, most state-of-the-art open source models use Rotary Embeddings. Rotary Embeddings are weak for length generalization. We show that combining Abacus Embeddings with RoPE does, in fact, yield improvement in operand length generalization. However, in Figure 7, we demonstrate the true potential for integrating Abacus Embeddings into a more general system, showing that the combination of Abacus Embeddings with FIRE unlocks generalization well beyond the problems that FIRE embeddings can solve on their own.

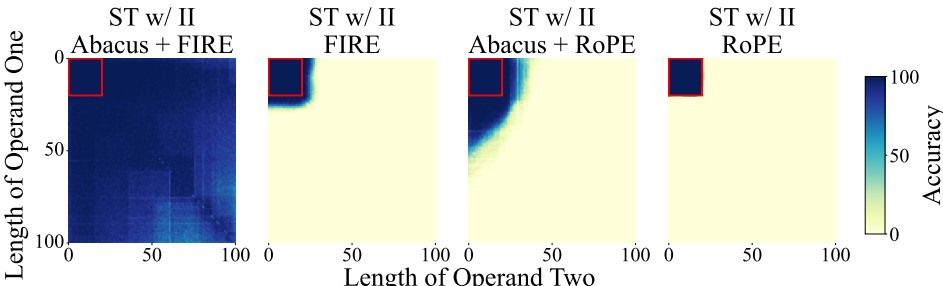

Figure 7: Exact match accuracy of standard transformer of depth 16 with input injection, trained on up to size 20 data. The red square denotes in distribution testing. Combining Abacus Embeddings with FIRE or RoPE embeddings improves out of distribution accuracy for addition, over the baseline models without Abacus Embeddings.

## 5   Discussion & Limitations

While the capabilities of LLMs have advanced far enough to encompass complex tasks including code generation and mathematical reasoning, stress testing the limits of these models remains a challenge. In this paper, we study mathematical reasoning tasks including addition, multiplication, and sorting to evaluate these capabilities in a controlled setting. We analyze the ability of specialized language models to learn algorithmic tasks in a zero shot setting, without access to outside tools like code interpreters, etc., exploring the benefits of various architectural improvements like improved embeddings and recurrent layers.

Across our experiments, we find that our novel Abacus Embeddings improve performance dramatically both when applied to standard transformers as well as recurrent variants. We repeatedly achieve length generalizations of at least $6\times$ (capped by the context length) more than doubling the extrapolation demonstrations in prior work, achieving near perfect results on addition of up to 100 digits, with repeatable results across multiple training runs. We demonstrate the the complementary properties of our Abacus Embeddings with other relative embeddings like FIRE, achieving dramatic

improvements in in-distribution multiplication performance, and making headway on the challenging problem of variable length array sorting.

Contrasting with prior work, our experiments explore types of extrapolation well beyond just length generalization for addition, presenting an architecture modification that improves performance on multiple algorithmic reasoning tasks simultaneously. We hope that our work deepens the community's understanding of these problems and paves the way for further advancements in the algorithmic reasoning capabilities of large language models.

**Limitations** There are some intrinsic limitations that accompany any study involving language model training from scratch under compute constraints. However, the primary point of relevance for this study is that although we show the compatibility of Abacus Embeddings with FIRE and RoPE embeddings, we do not actually explore any natural language tasks. In the future, a larger scale study including natural language would be needed to understand further how Abacus Embeddings would perform on heterogeneous tasks comprising both numerical and natural language inputs.

## Acknowledgments and Disclosure of Funding

This work was made possible by the ONR MURI program and the AFOSR MURI program. Commercial support was provided by Capital One Bank, the Amazon Research Award program, and Open Philanthropy. Further support was provided by the National Science Foundation (IIS-2212182), and by the NSF TRAILS Institute (2229885). Computing resources were furnished by the Department of Energy INCITE Allocation Program, and Lawrence Livermore National Labs.

Furthermore, this work was performed under the auspices of the U.S. Department of Energy by Lawrence Livermore National Laboratory under Contract DE-AC52-07NA27344 and was supported by the LLNL-LDRD Program under Project No. 24-ERD-010 (LLNL-CONF-2000175).

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

# A  Appendix

**Author Contributions**

**Sean McLeish\*** – Led the project, developed the idea, contributed to code, organized majority of experiments and contributed to writing.
**Arpit Bansal\*** – Contributed large amount to the idea, contributed to code, organized experiments for sorting arrays, and contributed to writing.
**Alex Stein** – Contributed to code, and helped plan the experiments.
**Neel Jain** – Contributed large amount to writing, and helped plan the experiments.
**John Kirchenbauer** – Contributed large amount to writing.
**Brian R. Bartoldson, Bhavya Kailkhura** – Helped set up large scale parallel addition evaluations, contributed to writing.
**Abhinav Bhatele** – Contributed to writing.
**Jonas Geiping, Avi Schwarzschild, Tom Goldstein** – Developed the idea, helped plan and organize the experiments, and contributed large amount to the writing.

## A.1  Extended Related Works

### A.1.1  Positional Embeddings.

FIRE embeddings are additive embeddings in the attention mechanism: $A_{RPE}(X) = XW_Q(XW_K)^T + B$ where $B_{i,j} = f_\theta\left(\frac{\log(c(i-j)+1)}{\log(c\max(i,L)+1)}\right)$ and $c, L$ are learnable parameters. Li et al. [2023] show empirically that these embeddings allow for length generalization and theoretically show they are capable of representing many other embedding types. Ruoss et al. [2023] propose using a random subset of a larger set of possible positions during training so that larger positional embeddings are trained. Zhou et al. [2024] use randomized FIRE [Ruoss et al., 2023, Li et al., 2023] embeddings to achieve length generalization on arithmetic tasks, which use randomized positions as input to the small multi layer perceptron used in FIRE embeddings.

## A.2  Datasets

**Addition:**  We sample equally, with replacement, from all $i \times i$ possible operand lengths up to the maximum dataset size of 20 million, we call this a dataset of size $i$ in the main text. For evaluation we sample 100 samples for each pair of operand lengths evaluated.

**Bitwise OR:**  The input for this problem is two binary vectors, the longer input vector is all zeros and the shorter input contains a one. The output should be the length of the longer vector with the one in the same position as in the shorter vector. If the inputs are the same length, the one can be in either vector. E.g. $001 \oplus 00000 = 00100$. For training, we exhaustively sample the space of all vectors of sizes less than or equal to the predefined maximum input vector size.

**Sorting:**  Given a list of reversed integers indexed by characters, output the characters in ascending order. E.g. $a : 64957, b : 99963, c : 10218, d : 7141, e : 05781 = d, e, b, a, c$. We implement the sampling process for sorting in a grid like manor. We query each "square" of an $[1, n] \times [1, n]$ grid until the maximum size has been reached for the dataset. When querying "square" $(i, j)$ we randomly sample $i$ integers of size less than or equal to $j$ digits. We randomly sample consecutive indices for the natural numbers in our list at both train and test time.

**Multiplication:**  We implement the multiplication datasets for both training and testing the exact same manor as for addition, only changing the operation used to calculate the answer.

## A.3  Bitwise OR on Binary Vectors

A necessary condition to perform addition is aligning digits of the same significance. We begin by examining positional embeddings for exactly this task. To do this we analyze the bitwise OR task, where the model has to output left aligned position wise OR of two binary vectors. We present samples from the dataset in Section A.3.1, these are left aligned to be representative of the task of aligning digits for reversed addition.

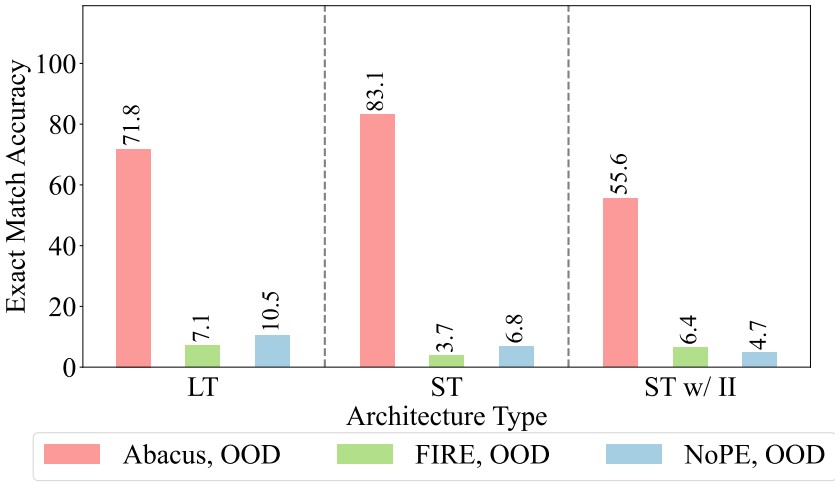

Figure 8: Accuracy of models on the bitwise OR task when trained on data with size up to 20, varying over different positional embeddings and architectures. Abacus Embeddings heavily improve performance on this task.

We train standard transformer, standard transformer with input injection and looped transformer models on the position wise or task, on a dataset where the maximum length of either input vector is twenty. This result is shown in Figure 8. Here we see that the Abacus Embeddings allow all models to generalize further on this task than the other embeddings which prior work for addition focuses on. As with addition, we see that looped transformers perform better than the standard architectures with FIRE or NoPE embeddings. We do note that these accuracies are not as high we report for addition. We hypothesize this is because the model is having to repeatedly predict the same token multiple times, this has been thought to be the cause of errors in prior addition work[Qian et al., 2022]. When we analyzed the errors in this task we found they were predominantly caused by the model outputting one too few or too many zeros.

### A.3.1 Example Data

$$000010 \oplus 00000000000000 = 00001000000000$$
$$000100 \oplus 0000000 = 0001000$$
$$001 \oplus 00000 = 00100$$

### A.4 Addition Models Trained on Varying Data Sizes

Across Figure 9, we see that increasing the size of the operands in the training set allows for better generalization above one hundred digits for all models. This is partially due to the sampling method for training Abacus Embeddings. As the offset randomization hyperparameter $k = 100$ is fixed across experiments, there are more embeddings trained if the operands seen during training are longer. The size of the OOD set below 100 is reduced as the size of the operands seen during training increases, as the ID category now includes this data. However, this does still show that the size of the operands seen during training directly impacts the generalization, with larger training sizes allowing for better generalization.

### A.5 Extreme Length Generalization for Addition

Absolute positional embeddings must be learned during training otherwise they are unusable at test time. This limits our Abacus Embeddings which are trained with the offset randomization hyperparameter $k = 100$. One possible way to resolve this generalization problem is to increase the value of $k$ during testing. In Figure 10 (left), we show the exact match accuracy of five looped transformer models, with eight layers in the recurrent block and two recurrences trained on size 20

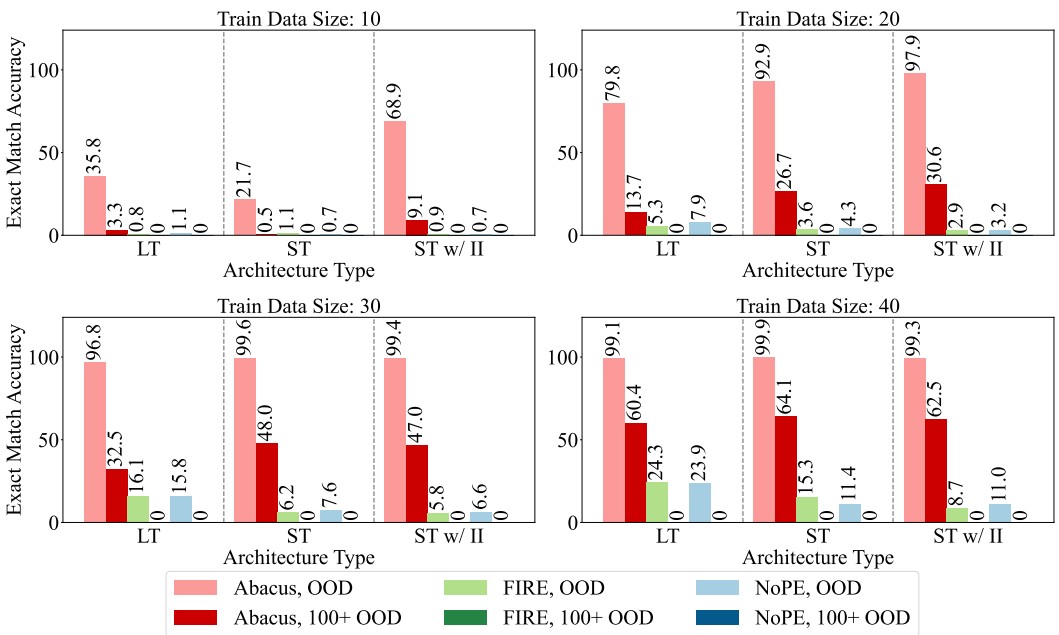

Figure 9: Mean exact match accuracy of three models of effective depth sixteen, varying the training data and architecture. We omit from the plot the in distribution accuracies as these are all 100% or very close to 100% for all models, this can be verified by the dark blue inside of all of the red squares in Section A.6. Models trained on larger operands achieve higher OOD accuracy.

data with Abacus Embeddings and $k = 101$, generalizing to 120 digit addition. We only show the accuracy for operands of the same length in Figure 10 (left), seeing these models consistently achieve accuracies of 95% and above. We see this across the paper this method is much more robust than that presented by Zhou et al. [2024].

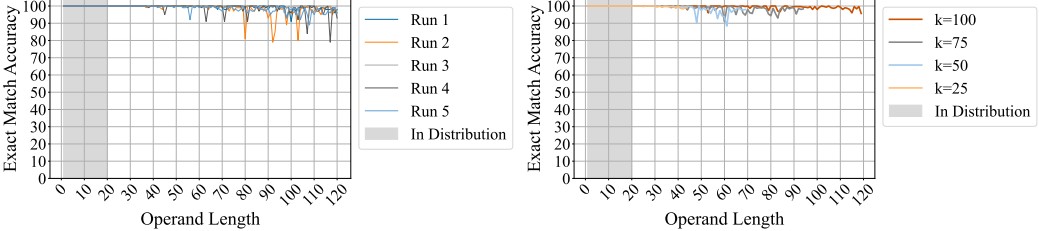

Figure 10: **Left:** Exact match accuracy of five models trained on size 20 data, generalizing well to 120 digit addition, an extrapolation of $6\times$. **Right:** Exact match accuracy of five models trained on size 20 data, offset randomization hyperparameter $k = 25, 50, 75$ and 100.
Only showing the accuracy for operands of the same length.

In Figures 10 (right) and 11 we continue varying the maximal offset randomization hyperparameter and size of the numbers in the training data. In Figure 10 (right), we show that varying the maximal offset randomization hyperparameter (k) changes the amount of extrapolation as we increase k to 100, as expected, This allows us to generalize to operands over a googol. In Figure 11 we show models trained on size 30 and 40 data with larger values for k, a maximum $6.8\times$ length generalization from training. We see the models struggle to use the largest embeddings, e.g. embedding 214 in Figure 11 (right), this is due to the stochastic training of embeddings, meaning the very largest embeddings are updated infrequently. This can be remedied by longer training but to remain consistent with other results we only train for 24 hours on a single A4000. Hence, we can easily increase k to larger values and perform arithmetic with far more digits, with suitable training data.

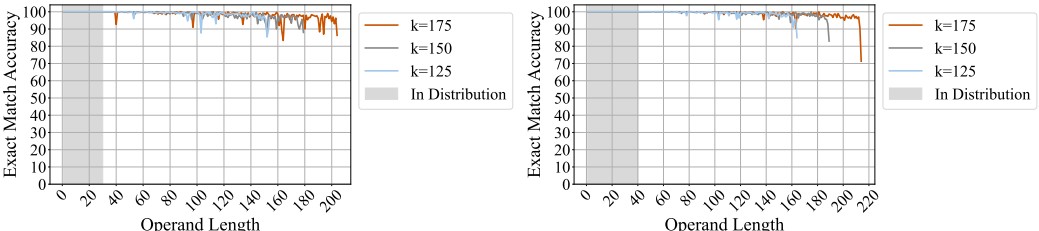

Figure 11: **Left:** Exact match accuracy of five models trained on size 30 data, offset randomization hyperparameter $k = 125, 150$ and 175. **Right:** Exact match accuracy of five models trained on size 40 data, offset randomization hyperparameter $k = 125, 150$ and 175.
Only showing the accuracy for operands of the same length. These results are from models which have 8 layers in recurrent block and 2 recurrences and are trained on size 30 data with varying k, each line is the average of 3 models.

### A.6 Addition Full 100 x 100 Plots

Here we present the mean accuracy as heatmaps for the main addition experiments shown throughout the paper. Figure 12 (left) corresponds to Top Left of Figure 9. Figure 12 (right) corresponds to Top Right of Figure 9 and Left of Figure 3. Figure 13 (left) corresponds to Bottom Left Figure 9. Figure 13 (right) corresponds to Bottom Right Figure 9 and Right of Figure 3. Figure 14 corresponds to Figures 4 and 17. All of these figures show the Abacus Embeddings ability to generalize in both dimensions of the addition problem.

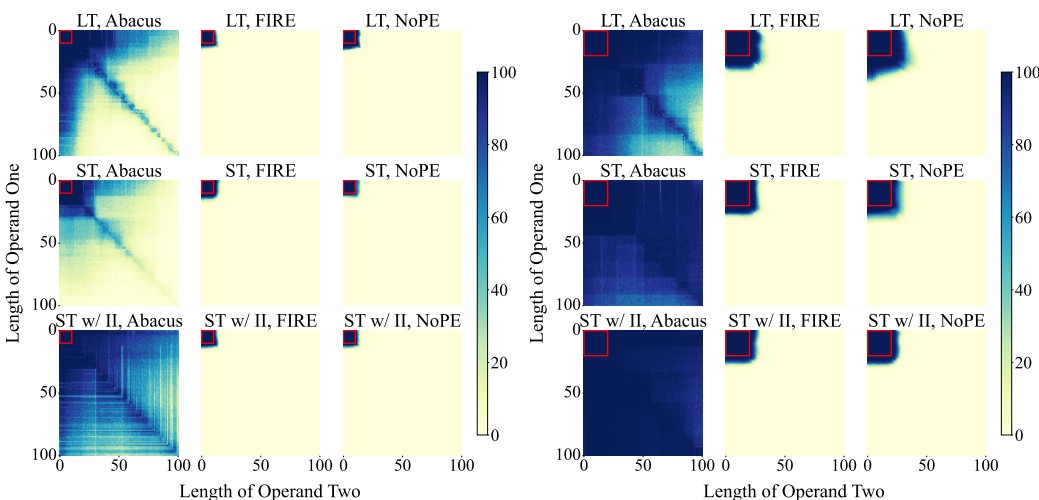

Figure 12: Full $100 \times 100$ exact match accuracy plots, taking the mean over three models. **Left:** Size 10 training data, corresponding to Top Left of Figure 9; **Right:** Size 20 training data, corresponding to Top Right of Figure 9 and Left of Figure 3.

### A.7 Addition Ablations

#### A.7.1 Analyzing the Intermediate Properties of Recurrence

Thanks to the looped transformer architecture, we can extract intermediate solutions from the models, allowing us to plot the models outputs over iterations of the recurrent block. We present an example in Figure 15 and suggest that this level of interpretability could be leveraged in future work. The model presented is a $1 \times 16$ model, one decoder layer and sixteen recurrences. We do not show the full 16 iterations in this plot for readability but these models do maintain a fixed point to 16 iterations and beyond.

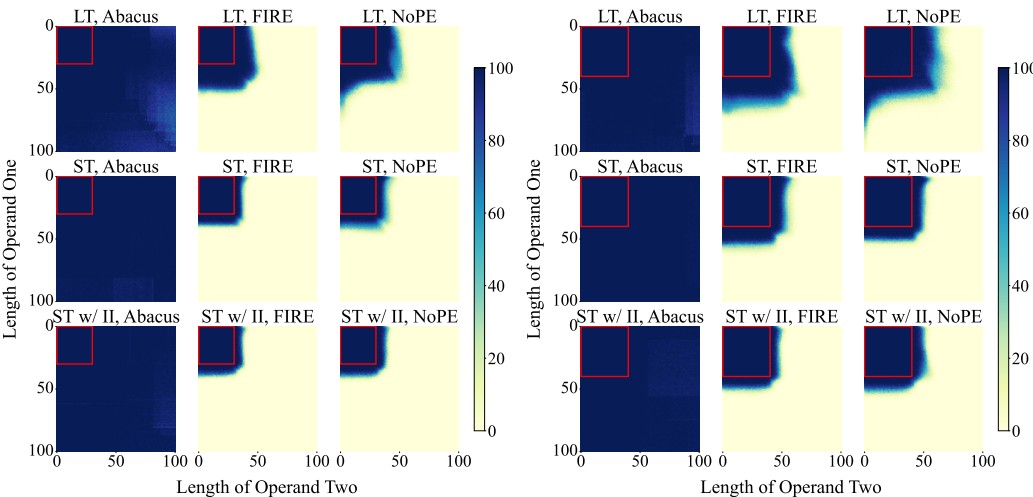

Figure 13: Full $100 \times 100$ exact match accuracy plots, taking the mean over three models. **Left:** Size 30 training data, corresponding to Bottom Left Figure 9; **Right:** Size 40 training data, corresponding to Bottom Right Figure 9 and Right of Figure 3.

### A.7.2 Removing Masking Before Equals

We mask all tokens before the equals sign in all of our experiments, we hypothesize that with more training time this constraint may be able to be removed. In Figure 16, we show the effect of training with the same amount of flops as the other addition experiments without masking before the equals sign.

### A.7.3 Varying Effective Depth

We begin in Figure 17 by showing a replica of Figure 4, this time including comparisons to FIRE and NoPE embeddings. Seeing, yet again, the improvements Abacus Embeddings give for addition.

In Figure 18, we present models with effective depths 8 and more than 16, respectively. In Figure 18 (left), we see that the effective depth 8 models under perform the models with 8 layers in the recurrent block and two recurrences shown in Figure 4, demonstrating the benefit of recurrence in this case. We see very high accuracy from all models in Figure 18 (right). Again, the depth 32 recurrent models outperform the standard models with input injection, even though it only has approximately a quarter of the parameters and achieves the highest OOD mean accuracy of all models presented. These ablations show that with Abacus Embeddings the addition task can be learned across many effective depths to varying degrees of accuracy.

In Figure 19 (left), we remove the input injection to the intermediate layers in the recurrent block, only keeping input injection to the first layer of the recurrent block. In Figure 19 (right) we divide the gradients in the recurrent block by the number of recurrences for the looped transformer models during training. We see very minor performance changes for all models shown in Figure 19, with the $2 \times 8$ model improving its performance slightly in left plot and the $4 \times 4$ model improving slightly in the right plot. We ablate this design choices as we have to remove the input injection inside of the recurrent and divide the gradients in the recurrent block by the number of recurrences for the multiplication models show in Figure 6. Hence, we can conclude there would only be very minor performance changes in this case for addition.

### A.7.4 Adding randomized Padding

Abacus Embeddings give strong priors for numerical tasks but without them, looped transformers perform better than the standard transformer architectures we present. The result shown in Figure 20 aligns well with the hypothesis that with fewer priors the looped transformer models are able to generalize better. In this case the priors are reduced as the training data is noised with random pad

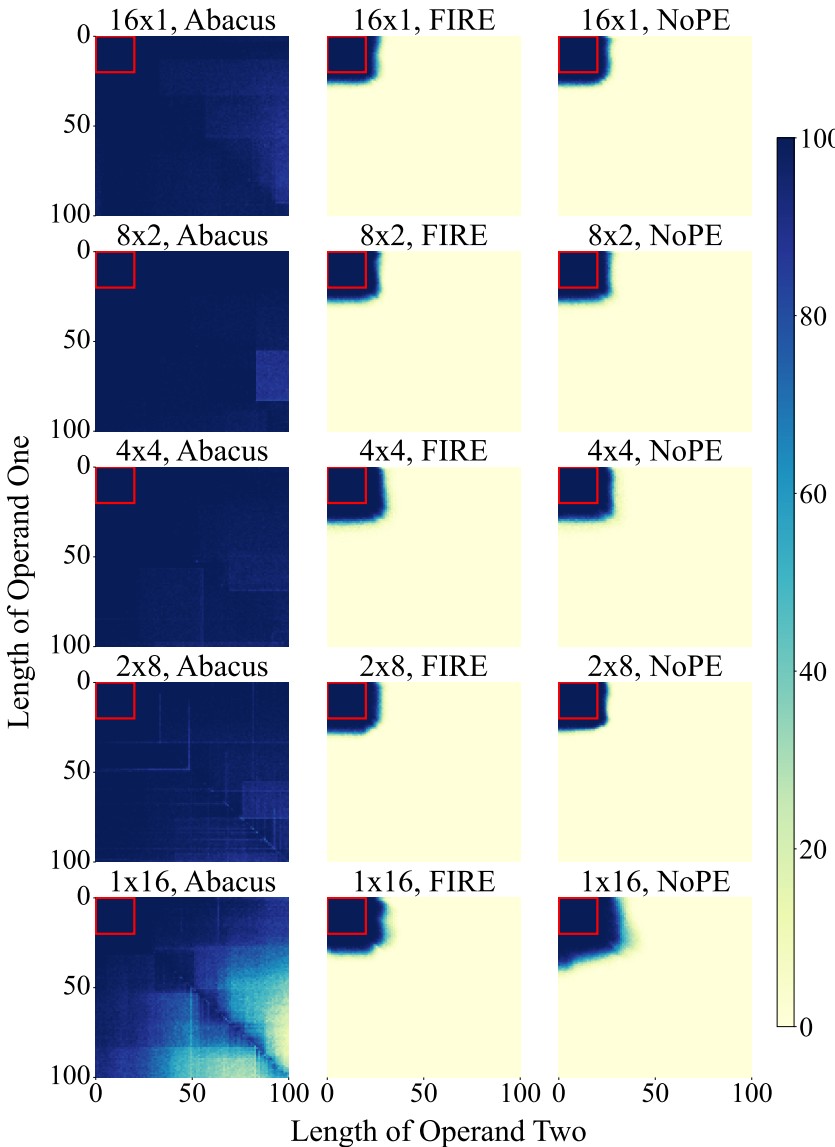

Figure 14: Full 100x100 exact match accuracy plots, taking the mean over three models, relating to Figures 4 and 17.

symbols, a method which was shown to improve length generalization in prior work [Shen et al., 2023].

### A.7.5 Index Hints

Zhou et al. [2023] "randomly sample consecutive index hints from a pre-defined ordered set of hints with 102 symbols," for example $a6b7c5 + a1b6c3 = a7b3c9$. We implement this method two ways. Firstly, cyclic, here we treat the list as cyclic when sampling. Secondly, non-cyclic, this reduces the number of samples which receive the embeddings later in the ordering as we only sample from the list in order. We see similar results for models trained on up to twenty digits as Zhou et al. [2023]. We do note that our format of taking the mean exact match accuracy does highlight robustness as if one of the three models tested were to not generalize well, this would impact reported accuracy heavily. We only show a comparison to size 20 training data due to the increased cost of evaluating these index hint models, as the inputs and outputs are approximately double the length of regular questions the inference time is heavily increased. Due to the robustness issues highlighted by Zhou

## Intermediate Outputs Over Recurrences

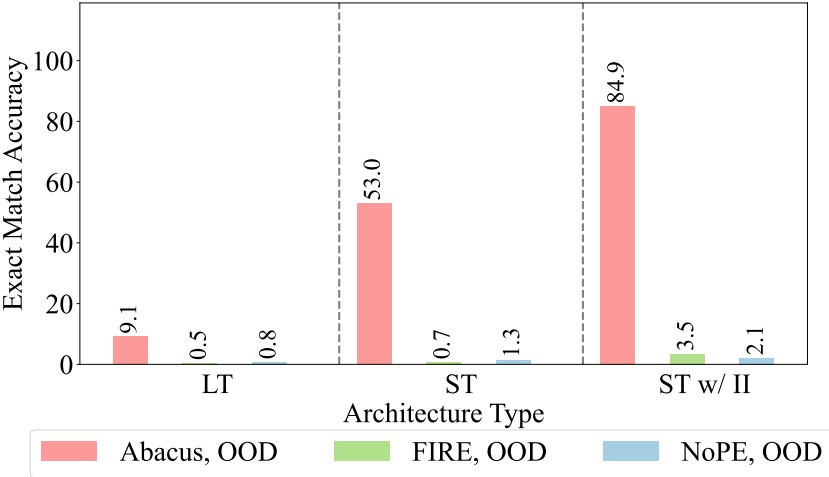

Figure 15: Plot showing the improvement of the prediction over "thinking" iterations on a 100 digit addition problem.
Input Prompt:
5879287854346790803556089719498716671892210129414436974968915190512644198885716176176
0096255295233702836+4358110391552830769683978480187501721764900525218097903808750
786159803668915002036143168815597779644=
Answer:
9195760736263745508459116846300200841916587728919941054185275957502629432039284417
58606474262584957001[EOS]
(Note that the plot is truncated.)

Figure 16: Effect of removing the masking of the loss before the "=" sign in the addition task. All models perform worse when trained for 24 hours on a single Nvidia RTXA4000 if we do not mask the input question in the loss function.

et al. [2024] with their methods, we try to the best of our abilities to faithfully reproduce their work within our experimental set up, noting that perhaps a better random seed or initialization may be able to produce better results for these models.

### A.8    Additional Experimental Information

In this work, we consider three different model types, the classical standard transformer, standard transformer with input injection, and looped transformers. We visually describe these in Figure 22.

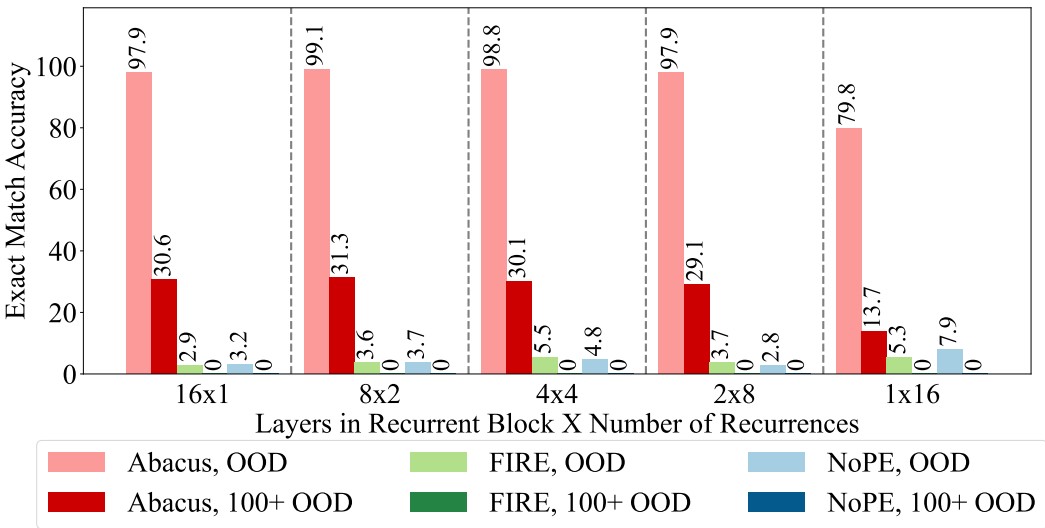

Figure 17: Continuation of Figure 4, including FIRE and NoPE embeddings. We see the Abacus Embeddings perform best for all models.

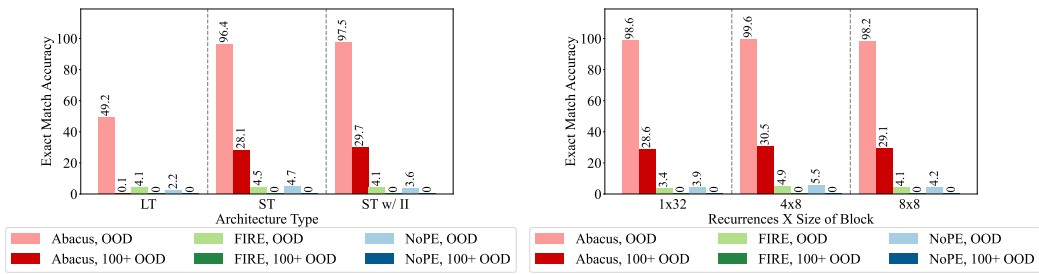

Figure 18: **Left:** Effective depth 8 models, trained on size 20 data. These models under perform the models with eight layers in the recurrent block and two recurrences shown in Figure 4, showing the benefit of recurrence for addition. **Right:** Effective depth >16 models, trained on size 20 data. The models contain many more parameters than all other models we present, showing more that an effective depth of more than 16 does not necessarily improve accuracy in this setting.

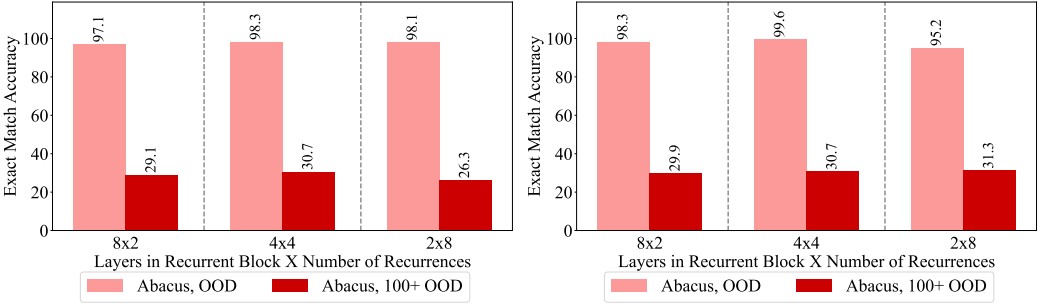

Figure 19: Replicas of the looped transformer models shown in Figure 4, to check the modifications we use to train addition models do not adversarially impact addition training, taking the mean of three models in each case. **Left:** without the input injection to the layers inside of the recurrent block, only to the first layer of the recurrent block. **Right:** dividing the gradients in the recurrent block by the number of recurrences.

Due to the looped transformer architecture the number of recurrences at train time can be different to the number of recurrences at test time, although we do not make use of this in this work.

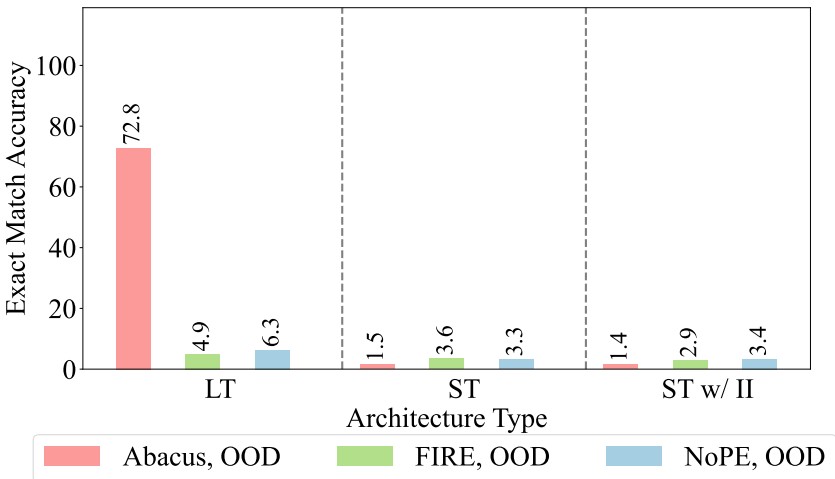

Figure 20: Effect of adding randomized padding into training data only for the addition task. Looped transformer models are able to maintain high accuracy when random padding is added into the data.

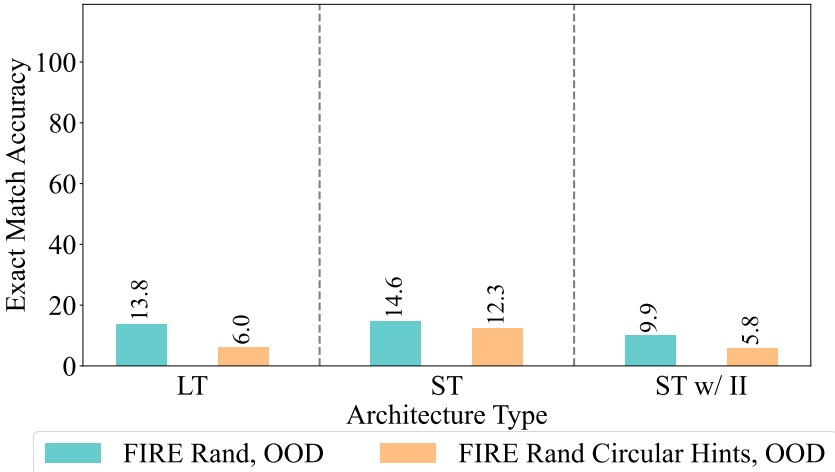

Figure 21: Using index hints and randomized FIRE embeddings, presented by Zhou et al. [2024], training on size 20 data with our methodology, such as masking before the equals sign. This would be comparable to "1 to 20" in Figure 13 presented by Zhou et al. [2024] and Figure 3 of our work.

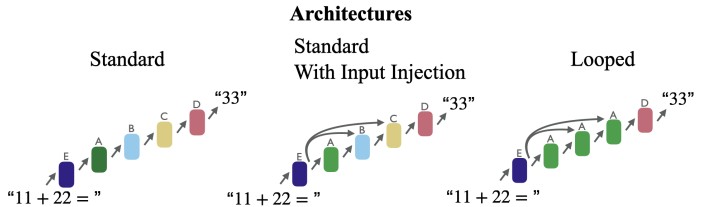

Figure 22: Visualization of the three architectures we study.

As Abacus Embeddings are a variant of absolute embeddings, reused only for numbers, they could be combined with relative embeddings being deployed in current models. If all digits input to the model are tokenized individually, we can perform a linear time operation to find and assign relative embeddings to all numbers in an input, which is lower than the quadratic cost incurred by attention. Training a small number of Abacus Embeddings may be enough to handle all numerical inputs for

Table 3: Number of parameters, to the nearest million, in a model with Abacus Embeddings and input injection.

| Layers in Recurrent Block | Recurrences | Parameters (Millions) |
|:---:|:---:|:---:|
| 16 | 1 | 122 |
| 8 | 2 | 64 |
| 4 | 4 | 34 |
| 2 | 8 | 19 |
| 1 | 16 | 12 |

Table 4: Default number of Nvidia GPU hours used to train a model.

| Dataset | Number of GPU Hours (training) | Number of GPU Hours (testing) |
|:---|:---:|:---:|
| Addition | 24 - RTXA4000 | 65.8 - V100 |
| Bitwise OR | 1 - RTXA4000 | 45 - V100 |
| Sorting | 24 - RTXA4000 | 64 - RTXA4000 |
| Multiplication | 192 - RTXA4000 | 0.83 - RTXA4000 |

addition as they are reused. To fully implement our methodology all numbers also have to be reversed, this can be implemented with simple regular expressions on all inputs and outputs.

We use a character level tokenizer for all experiments and greedy decoding in all testing. We train all models with a local batch size which is the maximum batch size that is a power of two that will fit into the sixteen gigabytes of GPU memory. For multiplication models we first take the mean loss across samples before taking the mean across all samples in a batch, instead of taking the mean loss across all token in a batch; we find this leads to slightly more stable training. We note that training models to solve multiplication requires more hyperparameter tuning than addition, perhaps implying it is a trickier task to learn. Also, FIRE models require a much greater compute budget for hyperparameter search as compared to Abacus models for multiplication. In Table 3, we present the approximate parameter counts for models trained with input injection and Abacus Embeddings.

**Compute Usage.** We detail the default use of GPUs for each experiment in Table 4. For some experiments, such as extreme length generalization (Figure 10) and index hints (Figure 21) more GPU hours are required for testing, these are included in the total number of GPU hours used. Our testing pipeline for addition and Bitise OR uses Nvidia V100 GPUs. Due to a technical problem, 'torch.compile' cannot be used on the V100 GPUs we use, therefore others may be able to reduce this compute time in future studies. All compute was provided by internal resources. During the exploratory phase of this project, we used more GPU hours to test and design the experiments shown, using approximately 1.5 terabytes of storage of the entire project. An estimate of the total compute required for all of the results presented in the main paper is $10,039$ GPU hours. The appendix results require a further $18,278$ GPU hours.

### A.8.1   Hyperparameters

We detail what we believe to be an important subset of the default hyperparameter values in Table 5. A full list of all hyperparameters and model configurations is contained in the code release. For multiplication models with FIRE embeddings, the learning rate is 0.00006, due to large instabilities in higher learning rates which were not experienced for the Abacus Embeddings.

### A.8.2   Code Release

We will release all code and datasets on GitHub with an MIT License.

Table 5: Default hyperparameter values.

| Hyperparameter | Default Value |
|---|---|
| Hidden Size | 1024 |
| Intermediate Size | 2048 |
| Embedding Size | 1024 |
| Number of Attention Heads | 16 |
| Progressive Loss Alpha [Bansal et al., 2022] | 1.0 |
| Data Type | float16/float32 |
| Optimizer | AdamW [Loshchilov and Hutter, 2017] |
| Global Batch Size | 8192 |
| Batch Size Ramp | 0.6 |
| Learning Rate | 0.0001 |
| Learning Rate Scheduler | Trapezoid [Zhai et al., 2022] |
| Activation Function | GELUglu [Shazeer, 2020] |
| Normalization Layer | LayerNorm [Ba et al., 2016] |
| Normalization Type | Post |
| Offset Randomization Hyperparameter ($k$) | 100 |
| Initialization | Deepnorm [Wang et al., 2022] |

