# OpenReview forum: "Transformers Can Do Arithmetic with the Right Embeddings"
_NeurIPS.cc/2024/Conference — NeurIPS 2024 poster_

### Official Review · Reviewer_GBbn · 2024-07-09

**Soundness:** 2
**Presentation:** 3
**Contribution:** 2
**Rating:** 6
**Confidence:** 4

**Summary:**

This paper introduces a simple yet effective encoding scheme that can be used to address the limitations of transformers at representing positional information, which is crucial in many algorithmic tasks such as those involving arithmetic operations. The authors propose an ad-hoc positional embedding, called “abacus embedding”, which encodes the location of each digit relative to the start of the current number and thus provides an explicit signal that the transformer can use to align digits. The effectiveness of the method is tested on addition, multiplication and sorting problems, with a particular focus on out-of-distribution test cases.

**Strengths:**

I think that this work is interesting and relevant. Although addition, multiplication and sorting problems might be considered trivial test cases because they can easily be solved with symbolic algorithms, they constitute an important benchmark to evaluate the algorithmic reasoning skills of neural networks, as also attested by the increasing interest of the deep learning community on mathematical tasks. The paper is well-written, and the method is clearly presented. The generalization achieved in the addition task is quite impressive, showing that the abacus embeddings enable a generalization factor of 6x in the OOD regimen. Although simple and straightforward, the proposed method seems original.

**Weaknesses:**

The abacus embeddings are defined according to the hyperparameter k, which is fixed a priori (e.g., k = 100). This limits the flexibility and generalizability of the proposed encoding scheme.

The authors deploy different architectures / hyperparameters to learn different problems (addition vs. multiplication vs. sorting). Since they argue that their architecture modification “improves performance on multiple algorithmic reasoning tasks simultaneously” it would be important to show that different tasks can really be learned simultaneously, without the need to build ad-hoc models for each algorithmic task that needs to be solved.

It is true that arithmetic operators are binary and thus “both addition and multiplication accept only two operands”. However, we can have a sequence of additions / multiplications, and it is well-known that also increasing the number of terms in arithmetic expressions causes troubles to transformers.

Because of these key issues, I think that the impact and significance of this work are not strong enough for a top-tier venue like NeurIPS.

**Questions:**

- How could we address the fact that the hyperparameter k needs to be fixed a priori?
- Can we implement a unified model that can learn all these tasks simultaneously?
- The proposed method achieves impressive OOD accuracy for addition, but only works “in distribution” for multiplication. It would be important to investigate this phenomenon more in depth.
- At least for addition, it would be useful to test OOD generalization by also adding more operands besides increasing the length of each operand.
- How does the present method compare to other recent proposals such as xVal (https://arxiv.org/abs/2310.02989)?
- I agree that addition, multiplication and sorting are good benchmarks because they are simple yet challenging; however the authors could better stress that these tasks are part of a broader class of elementary numerical skills that transformers struggle to learn (for a comprehensive review, see https://www.mdpi.com/2076-3417/14/2/744).

**Limitations:**

The authors properly addressed the limitations of their study.

---

> ### Author Rebuttal · Authors · 2024-08-06
>
> Thank you for your valuable time, your comments have greatly improved our draft.
>
> We answer your questions below in order:
> 1. While choosing k a priori could be a barrier to adoption of Abacus Embeddings in the community, we have shown that we can scale to at least one hundred more digits at test time than seen at training time for addition in our original submission. We believe that use cases with distribution shifts larger than this are more limited. Furthermore, in our general rebuttal response and in Rebuttal Figues 2 and 3, we show we can increase the distribution shift using a larger k and larger training data if a distribution shift of more than 100 digits is required.
>
>     For any language model a fixed context length is also decided a priori, in the extreme case one could choose an extremely large value of k, for example context_length/3, so that all two argument additions can be completed by model.
>
> 2. To see whether multi-skill training is possible, we trained a model on both addition and subtraction simultaneously without any hyperparameter changes from the addition models presented in the original draft. We show the results of this experiment in Rebuttal Figure 4, in the rebuttal pdf. Here we show that, even without hyperparameter tuning due to rebuttal time constraints, the models are able to extrapolate for both the symmetric addition operation and anti-symmetric subtraction operation simultaneously using a single model. We stress that we are training tiny transformers models in this paper, up to a maximum of 122M distinct parameters. We hypothesize that scaling either of these axes would allow for more simultaneous learning, however this is not the regime we analyze within this report.
>
> 3. We emphasize that we also achieve state-of-the-art performance on multiplication with Abacus Embeddings, when compared to prior work [1]. We agree that further improving performance for multiplication out of distribution is important future work and have updated the future directions section of our draft to emphasize this.
>
> 4. While this is a form of algorithmic generalization, we chose not to study generalization in the number of operands for addition or multiplication in this study, in a compute restricted regime. We instead highlight the generalization of operands in the sorting section.
> For sorting we have two dimensions of generalization. Firstly, “OOD in number length,” this is similar to addition and multiplication where we increase the number of digits in the numbers in the array being sorted. Secondly, “OOD in array length,” here we increase the length of the array so there are more numbers that need to be sorted. We analyze both of these dimensions in the reported “ALL OOD” accuracies where we scale each of these dimensions concurrently. Specifically, the “All OOD'' accuracies shown in Table 1 highlight that when both the number of operands and number of digits is varied during testing, the models trained with Abacus and FIRE Embeddings achieve the highest accuracy for the sorting task.
>
> 5. xVal, which embeds all real numbers by scaling a single fixed token-embedding, is a very important contribution to the language modeling community. We reference xVal in the Related Works section of our paper as it improves mathematical accuracy when the operands are small real numbers, up to 10e+8. However, it does not resolve the algorithmic generalization problems which we are working to solve in this paper, involving numbers much larger than 10e+8 (for example in Rebuttal Figure 3 we analyze up to 10e+214). The authors of the xVal paper highlight this in Section 4 - Future Directions, “Very large numbers saturate the normalization, as discussed in Sec. 2, and very small numbers are negligible from the model’s perspective.” This is because the authors “normalize numbers in the text corpus such that they fall within the range [−5, 5] as a preprocessing step before training.” Hence, we believe xVal would not be a compelling baseline for our study. We choose FIRE and RoPE as our main comparisons based on prior work and directly compare to the previous state-of-the-art (Randomised-FIRE with index hints) in Appendix Section: Addition Ablations - Index Hints.
>
> 6. We agree with and will act on this feedback for a future version of the paper, including the paper cited above, we have updated this in our draft.
>
> We believe the weaknesses are addressed in the reviewer questions. Should you have any further questions or require additional clarification, please do not hesitate to ask.
>
> [1]  Shen, Ruoqi, et al. "Positional description matters for transformers arithmetic." arXiv preprint arXiv:2311.14737 (2023).

---

> > ### Comment · Reviewer_GBbn · 2024-08-08
> >
> > I thank the Authors for having considered my comments. Having read their responses and the comments posted by the other Reviewers, I am persuaded to raise my score from 4 to 6.

---

### Official Review · Reviewer_mAhx · 2024-07-12

**Soundness:** 3
**Presentation:** 4
**Contribution:** 3
**Rating:** 6
**Confidence:** 3

**Summary:**

This paper studies a well-known problem, the length generalization issue of transformers in terms of doing arithmetic. This paper solves this problem via two natural strategies: (i) separate two operands via a newly proposed embeddings (Abacus Embeddings), and (ii) using looped Transformer architecture.

**Strengths:**

- The problem is well-motivated.
- The conjectures are very natural, and confirmed via extensive experiments.
- Experiments are well-designed and complete.
- The proposed solutions enjoy great performance.
- Considers diverse downstream tasks, including addition, multiplication, and sorting.

**Weaknesses:**

[Medium] The reason why looped transformer or recurrency helped with length generalization is still unclear, and in-depth analysis is needed. For instance, does the number of recurrence related to the length of the digits?

**Questions:**

Why does recurrency in terms of the model architecture help with length generalization?

**Limitations:**

See weakness.

---

> ### Author Rebuttal · Authors · 2024-08-06
>
> Thank you for your valuable time, your comments have greatly improved our draft.
>
> We answer your question below:
>
> We do not find that the number of recurrences is meaningfully linked to the length of the numbers in this study and we do a small visual analysis of the intermediate properties during recurrence in Appendix Figure 13. While we do find that a small amount of recurrence can lead to performance improvements throughout our work, we also show that we can achieve highly accurate out of distribution performance with standard decoder architectures using Abacus Embeddings.
>
> We hypothesize looped transformers may improve performance because they force the model to learn an algorithm that relies on an iterative process. This aligns their strategy with that of a human-designed algorithm, e.g. traditional addition algorithms repeat the same process iteratively for each pair of digits. The good inductive bias of recurrent models is more widely referred to as the theory of algorithmic alignment [1] and has motivated many algorithmic reasoning results, for example [2].
>
> We believe the weaknesses are addressed in the reviewer questions. Should you have any further questions or require additional clarification, please do not hesitate to ask.
>
> [1] Xu, Keyulu, et al. "What can neural networks reason about?" International Conference on Learning Representations (2020). https://openreview.net/forum?id=rJxbJeHFPS
>
> [2] Ibarz, Borja, et al. "A generalist neural algorithmic learner." Learning on graphs conference. PMLR, 2022. https://arxiv.org/pdf/2209.11142

---

### Official Review · Reviewer_uGwk · 2024-07-12

**Soundness:** 3
**Presentation:** 3
**Contribution:** 3
**Rating:** 7
**Confidence:** 4

**Summary:**

The paper studies the arithmetic capabilities of transformers and the problem of length generalization, specifically the ability to solve problems larger than the ones seen during training. It introduces Abacus Embeddings, a novel positional embedding that encodes the position of each digit relative to the start of the number. For multi-digit addition, Abacus Embeddings result in state-of-the-art generalization to sequences six times longer than the training sequences. Additionally, the paper explores the benefits of incorporating recurrent blocks, leading to further improvements. Finally, the paper demonstrates the effectiveness of Abacus beyond addition, showing success with in-distribution multiplication and array sorting tasks.

**Strengths:**

- **Originality:** While previous studies have noted that positional encoding can negatively impact the arithmetic generalization capabilities of transformer architectures, to the best of my knowledge, the introduced embeddings, the analyses, and the results presented in this paper are original.
- **Quality and clarity:** The work is technically sound. The experiments are well-designed and convincing, and the code for their implementation is provided in the supplementary material. The paper is clearly written.
- **Significance:** The goal of improving the extrapolation and reasoning abilities of transformers is both timely and significant. The results are very good, achieving state-of-the-art performance for length extrapolation in multi-digit addition.

**Weaknesses:**

1. The paper does not discuss the choice of the value used for the maximal offset randomization parameter $k$, which determines the distribution of the starting position of the first digit of the numbers during training. How was the value $k = 100$ chosen? Additionally, could higher values further improve extrapolation performance?
2. The sentence “our methods perform so well that we look beyond addition” (line 239) does not sound appropriate for a scientific paper. Please consider rephrasing it, for instance, “Given the strong performance of our method in the multi-digit addition task, we extend etc.”.

**Questions:**

3. Differently from addition, when considering multiplication, Abacus Embeddings achieve high in-distribution accuracy but struggle out-of-distribution (OOD), even when one operand is of unitary/short length. Do you have any insights about what might cause this significant difference? Do you have any ideas or potential modifications to the method that could improve OOD generalization in this context?
4. Could you elaborate more on how your embeddings could be integrated into settings that involve mixing arithmetic with natural language data?

**Limitations:**

The paper adequately discusses its limitations. I do not foresee any potential negative societal impacts arising from this study.

---

> ### Author Rebuttal · Authors · 2024-08-06
>
> Thank you for your valuable time, your comments have greatly improved our draft.
>
> We respond to your weaknesses and answer your questions below in order:
> 1. We find ~100 to be roughly optimal when the training data has a maximum of 20 digits; this already allows for addition of numbers larger than a googol. We describe and discuss the experimental evidence we gathered during the rebuttal period for varying the value of the maximal offset randomization hyperparameter (k) in the general rebuttal response and pdf. From this we conclude that the value of k can be varied with larger numbers in the training data allowing for larger values of k to be used.
>
> 2. We agree with the reviewer and have updated our draft.
>
> 3. While our results for multiplication are currently state-of-the-art compared to prior work [1], the reviewer is correct that we do not observe out-of-domain generalization. We do not know how to solve this problem at this time, but we do have several hypotheses on why do not see larger gains:
>     * Our multiplication models are small compared to the multi-billion parameter models we have become accustomed to in the open source community. With more compute or larger models, it may be possible to improve upon our results.  This direction is motivated by the observation that multiplication models, even with their increased compute budget compared to addition, do not achieve as low of a training loss as addition models. Furthermore, for addition models the loss plateaus at low values at the end of training, allowing for a period of training on low loss values. This does not occur for multiplication models, leading us to speculate that there is something to be gained from larger-scale training.
>     * Multiplication requires more abacus embeddings during training and testing, due to the increased output length.
>
>     We have updated our draft to include these as future directions of research.
>
> 4. Provided a model tokenizes numbers by their individual digits, which is done in most new open source models (e.g. llama and gemma), a plug in and play version of Abacus Embeddings (which is our sorting implementation in the supplementary material) can be used alongside other positional embeddings for language tasks. This code simply identifies all digit tokens in the input sequence and calculates the Abacus Embeddings in parallel for a batch, these can then be added to the input embeddings. Abacus Embeddings also rely on a least significant digit first format which can be done during encoding and decoding in the tokenizer with a simple regex to reverse all numbers.
>
> Should you have any further questions or require additional clarification, please do not hesitate to ask.
>
> [1]  Shen, Ruoqi, et al. "Positional description matters for transformers arithmetic." arXiv preprint arXiv:2311.14737 (2023).

---

> > ### Comment · Reviewer_uGwk · 2024-08-13
> >
> > Thank you for your response. I will keep my initial positive evaluation. Best wishes.

---

### Author Rebuttal · Authors · 2024-08-06

Our revised manuscript will include the following new experiments and discussion, which better clarify the broad utility and flexibility of Abacus Embeddings. We thank the reviewers for their questions and suggestions that led to these new positive results!


**Varying the value of the maximal offset randomization hyperparameter (k)**

In Rebuttal Figure 1, in the rebuttal pdf, we show models trained with k=25, 50, 75 and 100 trained on size 20 data; with the k=100 models being taken directly from Figure 4 of the current paper. We show the average of 3 models in each case analyzing accuracy only where both operands are the same length, similarly to Figure 9 of the current submission, to save on computation time during the rebuttal period. We see in the plot that these smaller values of k allow for good extrapolation and the amount of extrapolation depends on the value of k as expected. We find that increasing k to values more than 100 for models trained on data with a maximum of 20 digits, leads to diminishing returns within our experimental setup.

However, this can be resolved by including larger numbers in the training data. In Rebuttal Figures 2 and 3, in the rebuttal pdf, we show models trained on size 30 and size 40 addition data respectively. We show that larger values of k allow for much larger distribution shifts, with a maximum distribution shift of up to 175 digits being shown, this is a 6.8x length generalization from training to testing. Hence, we can easily increase k to larger values and perform arithmetic with far more digits than we did in our original state-of-the-art submission, with suitable training data.

**Learning addition and subtraction simultaneously**

In Rebuttal Figure 4, we train a model with exactly the same hyperparameters used to train the addition models in the submitted paper but this time also include subtraction examples in the data. We see that these small language models can simultaneously learn to extrapolate for both the symmetric operation of addition and the anti-symmetric operation of subtraction using Abacus Embeddings.

---

### Decision · Program_Chairs · 2024-09-25

**Decision:**

Accept (poster)

**Comment:**

The paper addresses the important and timely problem of length generalization in arithmetic tasks by introducing a simple yet effective embedding strategy that leverages the significance of each digit. This approach demonstrates impressive performance in solving length generalization challenges, showcasing the potential of straightforward methods in complex problem domains. This embedding also facilitates the use of looped TF models with input injection, resulting in a further improvement in length generalization accuracy.

The reviewers commend the paper for its clarity and quality of writing, which effectively conveys the concepts and results to the reader. The simplicity and effectiveness of the proposed method are particularly noteworthy. While the approach is somewhat tailored specifically to arithmetic tasks, this focus does not detract from its contribution, as it provides valuable insights that could inspire future research in related areas. Considering these strengths, the consensus is to accept the paper for publication.